# Discover-then-Name Revisited: Enhancing Concept Bottle-Neck Models Interpretability

## Abstract

This study replicates and extends the work on Discover-Then-Name Concept Bottleneck Models (DN-CBM) by Rao et al. (2024), which utilizes sparse autoencoders (SAEs) for automated concept discovery, enhancing traditional CBMs with improved concept generation and interpretability. Our replication of key experiments on CIFAR-10, CIFAR-100, Places365, and ImageNet confirms DN-CBM's automated concept discovery, task-agnostic applicability, and the generation of a more granular vocabulary. However, we find the claim of superior interpretability over CLIP to be inconclusive. Going beyond replication, we investigate the robustness of DN-CBM-discovered concepts. We analyze the variance in concept activation under color and reflection perturbations and employ Local Interpretable Model-Agnostic Explanations (LIME) to assess robustness against spurious correlations. Our novel experiments reveal a significant limitation in the model's robustness to color variations and spatial transformations. Furthermore, LIME analysis demonstrates that numerous concepts are unexpectedly activated due to spurious correlations. For meaningful concepts, we enhance interpretability by visualizing their corresponding image regions.

## 1 Introduction

Recent advances in deep learning have led to increased attention to the importance of explainability and interpretability, with mechanistic interpretability aiming to reveal how neural networks structurally process and represent information. The Concept-Bottleneck Model (CBM) promotes this by leveraging a set of human-interpretable concepts (e.g., "round," "yellow") to guide predictions; however, these concepts may not always be straightforward to detect or interpret. To address this, Rao et al. (2024) introduced Discover-Then-Name (DN-CBM), which uncovers the model's naturally learned concepts and assigns meaningful labels by aligning them with CLIP embeddings from a predefined vocabulary, thereby enabling a linear probe to classify images based on extracted concepts. We revisit DN-CBM in this work due to its promising approach to interpretable concept-based learning, particularly in evaluating its robustness—examining how reliably it discovers and leverages concepts under perturbations and distribution shifts.

In this work, we reproduce the author's experiments across CIFAR-10, CIFAR-100, Places365, and ImageNet to verify their claims and determine the robustness of DN-CBMs. We further investigate the SAE's color understanding by modifying the Places365 color map, assessing the impact of reflection transformations on concept generation, enhancing the interpretability of concept assignments, and identifying spurious correlations using LIME. Section 2 outlines the scope of our study, while Section 3 details the proposed methodology and experimental setup. Section 4.1 presents our replicated qualitative and quantitative results, followed by additional experiments and a survey study in Section 4.2. The reproduction process is examined in Section 5, with concluding remarks presented in Section 6.

## 2 Scope of reproducibility

This paper's scope of reproducibility focuses on the author's framework for reproducing the DN-CBM results. The main claims presented by the author are as follows:

1. **Automated Concept Discovery.** The paper introduces a method for automatically discovering semantically meaningful concepts from CLIP features using Sparse Autoencoders (SAEs), where semantically related concepts cluster together in the latent concept space.
2. **Enhanced interpretability.** Using SAEs leads to interpretable models by extracting and naming concepts that align well with human-understandable features, enhancing the model's transparency.
3. **Task-Agnostic Approach.** The feature extractor, the datasets used for concept discovery and downstream tasks, and the vocabulary used for naming concepts can be freely chosen.
4. **Impact of Vocabulary on Concept Name Granularity** The choice of vocabulary affects the granularity of concept name assignment. Expanding the vocabulary with more specific terms enhances the granularity of concept naming, while removing terms leads to less descriptive concept names.

In this work, we reproduce the experiments on all the downstream datasets to validate and verify the author's claims.

## 3 Methodology

### 3.1 CBM Construction

The CBM pipeline proposed by Rao et al. (2024) consists of 3 parts: automated concept discovery via sparse autoencoders (SAEs), automated concept naming by aligning the CLIP embeddings of predefined vocabulary with concept embeddings, and training the concept bottleneck model.

#### 3.1.1 Concept Extraction

First, the images are converted to the CLIP embedding space. The concepts are then discovered using the SAE, representing CLIP features in a high-dimensional space but with sparse activations to obtain interpretable concept representations (Bricken et al., 2023) (Cunningham et al., 2023). The SAE consists of a linear encoder $f(\cdot)$ with weights $W_E \in \mathbb{R}^{d \times h}$, a ReLU activation function $\phi$, and a linear decoder $g(\cdot)$ with weights $W_D \in \mathbb{R}^{h \times d}$. For a given input $a$, the SAE computes:

$$SAE(a) = (g \circ \phi \circ f)(a) = W_D^T \phi(W_E^T a) \tag{1}$$

The hidden representation $f(a)$ is designed to have significantly higher dimensionality ($h \gg d$) compared to the CLIP embedding space while optimized to activate very sparsely. Specifically, the SAE is trained with a reconstruction loss $L_2$, as well as a sparsity regularization $L_1$ on the CC3M dataset (Sharma et al., 2018):

$$\mathcal{L}_{SAE}(a) = \|SAE(a) - a\|_2^2 + \lambda_1 \|\phi(f(a))\|_1 \tag{2}$$

#### 3.1.2 Concept Naming

Once the SAE is trained, the latent representation produced by the encoder corresponds to the concept space where each concept can be named to extract human-understandable interpretations of the concepts.

To align concept representations with a set of words, the authors employ the 20k most commonly used English words from the CLIP-Dissect vocabulary (Oikarinen and Weng, 2023). The SAE decoder is designed to map a concept representation back to the CLIP embedding space using weights $W_D \in \mathbb{R}^{h \times d}$. In this context, a vector $p_c$ represents the concept $c$ in the latent space:

$$\mathbf{p}_c = [\mathbf{W}_D]_c \in \mathbb{R}^d \tag{3}$$

The SAE is trained to transform CLIP embeddings into concept representations through a reconstruction objective. This means that the SAE decoder can reconstruct the original CLIP embeddings from the concept representations produced by the encoder. Consequently, a specific concept representation $p_c$ can be decoded back into a CLIP embedding, which can then be compared to the CLIP embeddings of the CLIP-Dissect vocabulary. By calculating the cosine similarity between the decoded embedding and the embeddings of

the vocabulary words, we can associate the concept $p_c$ with the word $v_c$ from the vocabulary that has the highest similarity:

$$v_c = \arg\min_{v \in \mathcal{V}} \cos\left(\mathbf{p}_c, \mathcal{T}(v)\right). \tag{4}$$

### 3.1.3 Concept Bottleneck Model

The authors use a linear model designed to operate on SAE concept activations to perform the classification. The DN-CBM predicts the class of an input image $x_i$ as follows:

$$t(x_i) = \underbrace{h}_{\text{Probe}} \circ \underbrace{\phi \circ f}_{\text{SAE}} \circ \underbrace{\mathcal{I}}_{\text{CLIP}}(x_i). \tag{5}$$

The linear probe is trained using cross-entropy (CE) and L1 loss to increase interpretability (Bricken et al., 2023).

$$\mathcal{L}_{\text{probe}}(x_i) = \text{CE}\left(t(x_i), y_i\right) + \lambda_2 \|\omega\|_1 \tag{6}$$

where $\lambda_2$ is $L_1$ sparsity coefficient, $y_i$ is the ground truth label, and $\mathcal{I}(\cdot)$ is the CLIP model.

### 3.2 Datasets

The authors utilize the CC3M dataset to train the SAE model, while other datasets are used to train a linear probe for downstream tasks.

- **CC3M** is a dataset comprising approximately 3.3 million image-caption pairs, primarily designed for training models for image-text alignment tasks (Sharma et al., 2018).
- **CIFAR-10** is a dataset of 60,000 32x32 images distributed over 10 classes, with 6,000 images per class (Krizhevsky et al., a).
- **CIFAR-100** is a dataset similar to CIFAR-10 but features 100 classes, each containing 600 images, offering a finer-grained classification challenge (Krizhevsky et al., b).
- **ImageNet** is a large-scale dataset with more than 1 million labeled images spanning 1,000 categories, widely used to benchmark image classification models (Deng et al., 2009).
- **Places365** is a scene recognition dataset comprising 365 categories. For this study, we utilize the *Places365-Standard* version, which contains 1.8 million images (Zhou et al., 2017).

### 3.3 Experimental setup and code

We used the provided GitHub repository to replicate the authors' experiments, implementing minor bug fixes and adding a missing script for computing normalized CLIP embeddings for the CLIP-Dissect vocabulary (Oikarinen and Weng, 2023). We also modified the code to enable image sampling based on cosine similarity with vocabulary embeddings, providing a baseline for assessing the SAE's interpretability.

To verify that automated concept discovery produces semantically meaningful concepts, we apply K-Means clustering over concept activations on Places365 (subsubsection 4.1.1). To assess classifier interpretability, we analyze DN-CBM's predictions by identifying top contributing concepts per prediction and conducting a survey comparing CLIP's top-activating images to DN-CBM concepts.

To assess the classifier's interpretability, we examine DN-CBM's predictions by analyzing the top contributing concepts for each prediction and evaluating whether those concepts are relevant to the corresponding image. Additionally, we conducted a survey to compare the top-activating images for each vocabulary token in CLIP to those for the corresponding DN-CBM concepts, illustrating how the discovered concepts align with or differ from CLIP's learned representations.

Color plays a dual role in model performance. While it can be beneficial for distinguishing objects like flowers or fruits, it can also introduce biases by leading to an over-reliance on irrelevant features, such as when differentiating between a car and a plane. Ideally, a robust model should exhibit invariance to intraclass color variations (e.g., a car can be red or blue). To evaluate whether our generated concepts capture color information, we performed stratified sampling on 7300 validation images from the Places365 dataset, creating

three versions: original color, grayscale, and inverted color map. We then extracted the top 20 concepts for each version and scanned for basic color terms, including "noir," to assess the model's overall understanding of color schemes (Figure 12). To evaluate color invariance, we analyzed the overlap of the top 10 concepts across these three variations. A concept was categorized as color-invariant if more than five of the top 10 overlapped across the variations and color-dependent if more than five were unique to the original color image. We further examined the actual labels and predicted classes for a qualitative assessment of color sensitivity.

Similarly, reflection-invariance helps evaluate the model's reliance on the relative spatial arrangement of objects within images. Flipping an image should not alter the final prediction, as the semantic content remains unchanged. To further assess the model's robustness and the stability of concept generation, we extended our analysis by applying horizontal and vertical reflections to the previously stratified validation sample and measuring classification accuracy on the modified data.

Finally, we extend the original work by integrating LIME (Local Interpretable Model-Agnostic Explanations) to gain a deeper understanding of the origins of the generated concepts. This is particularly beneficial for concepts that exhibit a weak or unclear relationship with the input image or arise due to spurious correlations. To this end, we adapt LIME to quantify the contribution of individual concepts rather than predicting label confidence. We utilize the Python library `lime` (Ribeiro et al., 2016) and configure the `LimeImageExplainer` with the following parameters: `top_labels=5`, `num_samples=5000`, and `batch_size=1`. For the `get_image_and_mask` function, which is responsible for generating explanation masks, we set `positive_only=True`, `num_features=3`, `hide_rest=False`, and `min_weight=0.0001`. All other hyperparameters are maintained at their default settings.

### 3.4 Computational requirements

We perform all experiments on an NVIDIA A100 GPU. We replicate the experimental setup provided by the author, which utilizes Python 3.10. Carbon emissions are included and calculated using the Machine Learning Impact calculator (Lacoste et al., 2019). We conducted our experiments using a private infrastructure with a carbon efficiency of 0.555 kgCO2eq/kWh (Moro and Lonza, 2018). We summarize the total compute and emissions in Table 1. We trained an SAE for 200 epochs using CC3M (Sharma et al., 2018). Subsequently, for each of the five probe datasets, we conducted zero-shot classification and trained linear probes on both CLIP features and the concepts generated by the trained SAE. Furthermore, we performed a data augmentation experiment by adjusting color schemes on the validation set of the Places365 dataset (Zhou et al., 2017). Finally, we assessed how various image features contribute to activations in top concepts during inference using LIME on approximately 100 images.

|  | Reproducing Original Experiments | | | | | Experiments Beyond Original Paper | | Total |
|---|---|---|---|---|---|---|---|---|
|  | CC3M | CIFAR-10 | CIFAR-100 | Places365 | ImageNet | Data Augmentations | LIME | |
| Total compute time (h) | 1.5 | 0.14 | 0.35 | 4.6 | 9.24 | 0.67 | 4 | **20.5** |
| kgCOeq Emissions | 0.21 | 0.02 | 0.05 | 0.64 | 1.28 | 0.09 | 0.56 | **2.85** |

Table 1: Total compute time and carbon emissions for different experiments.

## 4 Results

Our reproducibility study confirms the accuracy of the final two claims outlined in Section 2. The assertions regarding automated concept discovery and enhanced interpretability are generally valid; however, there are instances where they do not hold. In this section, we present supporting results and discuss situations where certain claims fail.

### 4.1 Results reproducing original paper

#### 4.1.1 Automated concept discovery

Similarly to the authors, we demonstrate that the resulting cluster concepts are semantically meaningful, as illustrated in Figure 1a. The first set of images represents agriculture and crop production related concepts. The second cluster of features is about water-related activities. The final set depicts abandoned or demolished locations.

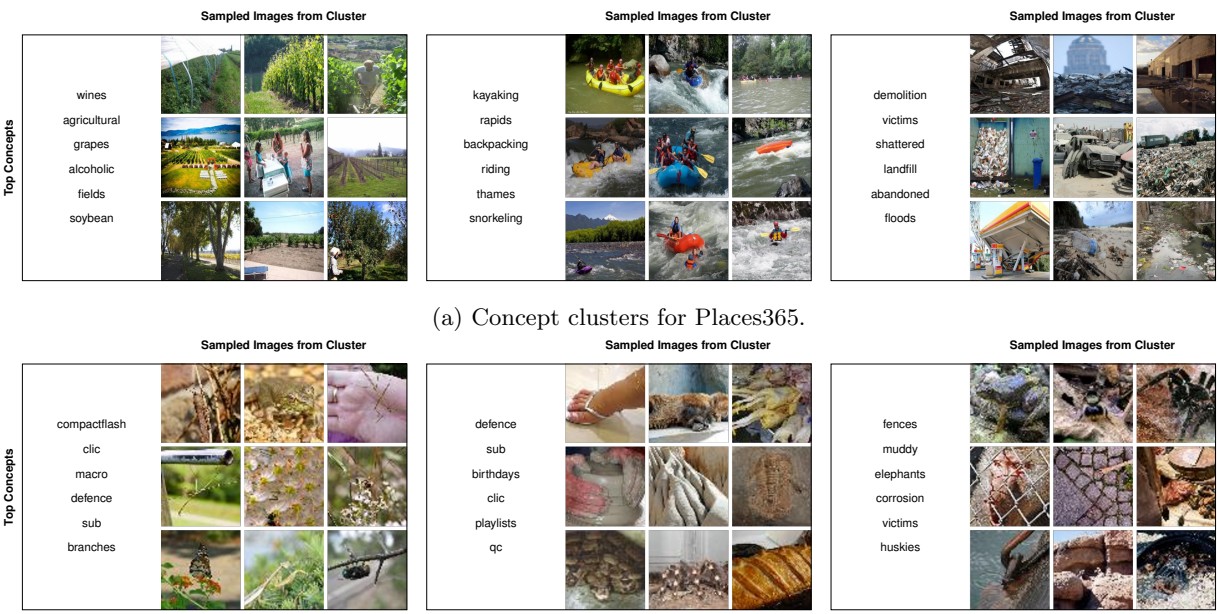

(a) Concept clusters for Places365.

(b) Concept clusters with non-descriptive names.

Figure 1: Comparison of concept clusters for Places365.

However, on numerous occasions, when clusters of images that could have been accurately described using appropriate words from the vocabulary were instead labeled with nonsensical in a given context words, as shown in Figure 1b. This problem frequently occurred with words assigned to multiple concepts. For instance, in the SAE we trained, 28 concepts were labeled with the word "clic", and 11 concepts were labeled with the word "sub."

Despite the described issues, most identified clusters are semantically consistent, demonstrating that the concepts discovered by SAE are effective for representation learning and are human interpretable.

#### 4.1.2 Enhanced interpretability of SAE generated concepts

The results for individual concept contributions show that classifier predictions are indeed human-interpretable. As illustrated in Figure 2, the top contributing concepts strongly align with both the actual content of the image and the predicted class.

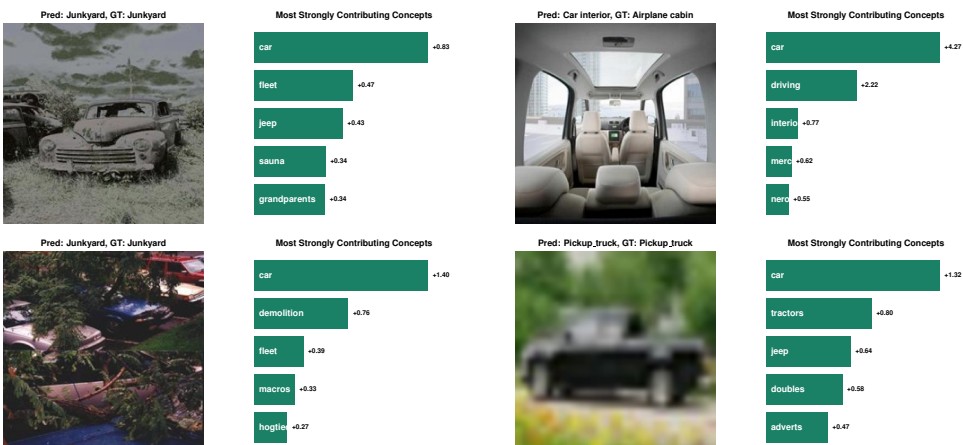

Figure 2: Local explanations for images related to cars.

Although most examples demonstrate that the primary contributing concepts align with the image content, seemingly meaningless concepts arise, as detailed in Appendix A.2. Generally, images whose content can be effectively described using terms from our vocabulary are well-represented by the extracted concepts. Conversely, cases where relevant concepts are present but the overall image description is poor are relatively infrequent. This suggests that DN-CBM, while often effective, is not inherently fully interpretable, as it occasionally fails to leverage relevant concepts when appropriate. Nevertheless, the main contributing concepts accurately reflect the visual content for most samples. To further assess the alignment between these contributing concepts and the objects in the image, we analyze the top concept contributions using LIME, as described in Section 4.2.3.

### 4.1.3 Task-Agnostic approach

| Model | ImageNet | Places365 | CIFAR-10 | CIFAR-100 |
|---|---|---|---|---|
| Linear Probe | 71.0 | 52.0 | 88.0 | 69.0 |
| *Reported* | *73.3* | *53.4* | *88.7* | *70.3* |
| Zero-Shot | 59.8 | 34.2 | 71.5 | 41.9 |
| *Reported* | *59.6* | *38.7* | *75.6* | *41.6* |
| DN-CBM | 71.0 | 50.0 | 89.0 | 69.0 |
| *Reported* | *72.9* | *53.5* | *87.6* | *67.5* |

Table 2: Top-1 Validation Accuracy(%) comparison of models across downstream datasets.

Rao et al. (2024) emphasize the agnostic approach of the DN-CBM framework. Essentially, the concept bottleneck layer is agnostic to the downstream dataset used for classification. The concepts discovered by the SAE on a given dataset, such as CC3M (Ng et al., 2021), allow for seamless application across various datasets for classification tasks. To verify this claim, we reproduce the classification experiments on the datasets mentioned in Section 3.2. As baselines, Rao et al. (2024) use a linear probe trained on the corresponding dataset's CLIP-RN50 (ResNet-50) image embeddings (Linear Probe) and CLIP-RN50's zero-shot classification performance (Zero-Shot).

Table 2 reports the top-1 validation accuracies for the baselines and DN-CBM. The linear probe trained on CLIP embeddings is similar to the reported scores. Similarly, the zero-shot performance of CLIP-RN50 broadly aligns with the reported baselines, aside from a minor ( 4%) shortfall on Places365, likely due to the absence of the exact CLIP prompt templates utilized by the author for this dataset. Our reproduced DN-CBM accuracies also closely match the reference values on ImageNet, Places365, CIFAR-10, and CIFAR-100, confirming the robustness of CBM approaches.

### 4.1.4   Impact of vocabulary on concept name granularity

The original authors argue that the granularity and size of the vocabulary play a crucial role in the accuracy of obtained concept names. They further suggest that vocabulary design can be leveraged to control the level of granularity in assigned names, depending on the specific use cases.

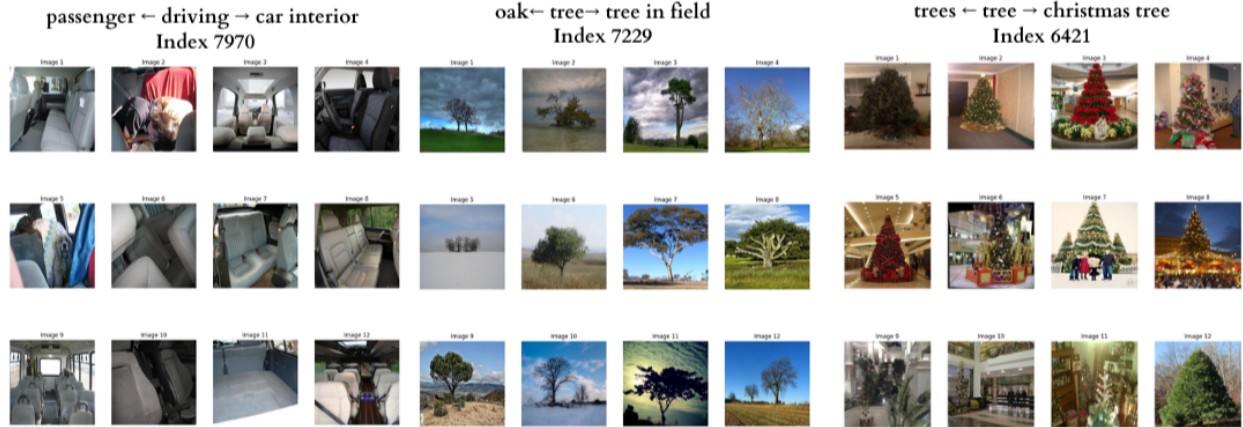

Figure 3: Impact of vocabulary on granularity of concept names

Our experimental findings confirm the validity of the author's claim that refining vocabulary enhances concept assignment accuracy. As illustrated in Fig. 3, our analysis of the concepts "tree" and "driving" demonstrates this effect. For instance, in the case of "driving," images depicting car interiors were more accurately labeled after adding the term "car interior" to the vocabulary. Conversely, removing "driving" reduced accuracy, leading to inaccurate assignments such as "passenger." Therefore, these results validate the authors' claim that a refined vocabulary improves the precision of the concept assignments.

### 4.1.5   Qualitative evaluation through a user-study

We surveyed 20 randomly selected concepts, categorized by alignment with text embeddings. For each concept, participants evaluated two image grids—one generated using CLIP features (Image 1) and one using DN-CBM features (Image 2)—ranking them on semantic consistency and name accuracy using a 5-point Likert scale. The survey received 22 responses. Additionally, a preference question asked users whether they preferred the images from CLIP, DN-CBM, or neither for each concept. Figure 4 visually summarizes the user ratings for semantic consistency and overall preference.

To assess statistical significance, we performed paired t-tests comparing the Likert scale ratings for semantic consistency and name accuracy between the two methods for each of the 20 concepts. The resulting p-values were combined using Fisher's method to determine overall significance. The combined p-values indicate a statistically significant difference between the two methods for both semantic consistency ($p \approx 7.41 \times 10^{-12}$) and name accuracy ($p \approx 5.93 \times 10^{-11}$) across all concepts evaluated.

While the overall differences are statistically significant, the visual representation of semantic consistency ratings in Figure 4a shows relatively small differences in the median and interquartile ranges, particularly for high and intermediate alignment concepts. This visual observation and variability in results across individual concepts might explain why the initial assessment suggested minimal differences, even though the aggregated statistical tests confirmed significance. This contrasts somewhat with claims by Rao et al. (2024) that DN-CBM clusters are perceived as definitively more semantically consistent and accurate. However, regarding user preference (Figure 4b), our findings align more closely with the original study; despite potential discrepancies in rated consistency or accuracy, participants generally favored the DN-CBM-generated images (Image 2) when asked which image set represented the word better for a given concept.

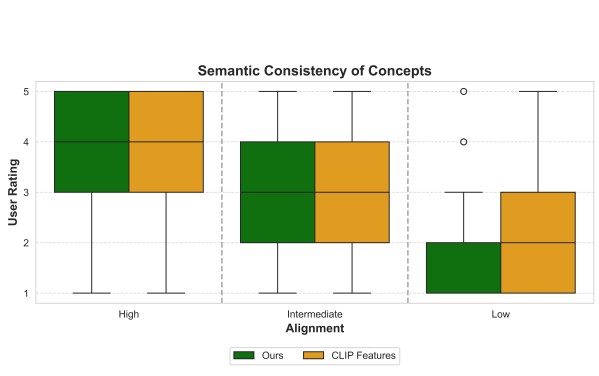

(a) Survey Results on Semantic Consistency

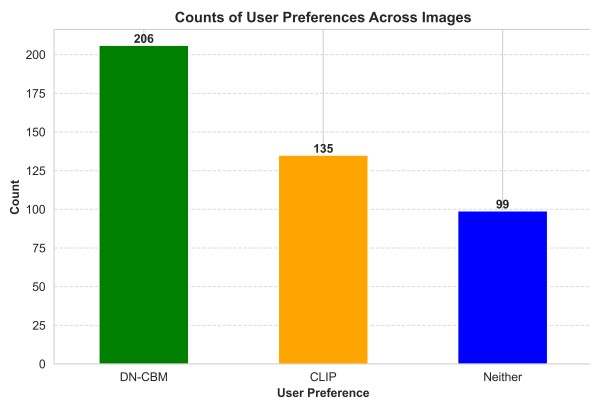

(b) Survey Results on User Preference

Figure 4: Comparison of survey results on semantic consistency (a) and user preference (b). The plots compare our method (DN-CBM) against CLIP features.

## 4.2 Results beyond original paper

### 4.2.1 Effect of color augmentation on concept representations

Understanding intra-class color variations is crucial for classifier robustness. If concept interpretation shifts significantly with color changes, it suggests a lack of robustness in the DN-CBM methodology. This experiment identifies the most affected classes by investigating concept and classification performance changes under varying image colors.

The first part evaluates color recognition through generated concepts. Observations suggest the SAE effectively captures different colors. As shown in Figure 12, the original image associates "yellow" with a bus, the grayscale version lacks color-related concepts except for "noir", and the inverted image correctly identifies "blue", matching the bus color. This hypothesis is further supported by the distribution of color-related concepts across different image variations (Figure 13). Results show that original images utilize a balanced color distribution, while grayscale images predominantly use "noir" and inverted images emphasize bright colors like purple and turquoise. This suggests the SAE accurately associates colors with images. Notably, color concepts appear more critical in augmented images than in original ones, likely because the unusual color schemes force the model to focus on color for accurate representation.

In the second part, we measured the change in the classifier performance depending on different color schemes. Results in Table 5 indicate a performance drop when colors are altered, with the inverted dataset showing the most significant decline (18.7% accuracy). Grayscale images also reduce accuracy, though less severely. This suggests the model relies on color-based concepts, with grayscale images simply removing color while inverted images introduce unnatural noise, disrupting the model's ability to interpret content effectively. Table 5 shows additional information on individual class performance. We observe the most significant accuracy drop for classes "swimming pool outdoor" (65% drop), "Garage outdoor" (65% drop), and "Legislative chamber" (65% drop). On the other hand, certain classes, such as "Cockpit" and "Boxing ring", maintain their accuracy despite the transformation. These results highlight how certain classes rely heavily on color (e.g., swimming pools often being blue), while others rely more on shape and texture-based features. Relying on color as a feature is not a bad thing on its own; however, for certain classes, such as "Garage Outdoor" it is likely to result in a far less robust classification during inference.

| Transformation | Original | Grayscale | Inverted |
| --- | --- | --- | --- |
| Accuracy (%) | 49.9 | 40.1 | 18.7 |

Table 3: Top-1 Validation Accuracy(%) comparison across transformations.

Finally, we explored color invariance in the generated concepts, focusing on two key research questions: (1) Which concepts remain consistent across different colors, and how frequently do they occur? (2) Which concepts are inherently dependent on color?

In our analysis, we found that instances of color independence were observed 366 times, accounting for approximately 5% of all images. A common characteristic of these images is their focus on a primary subject with minimal competing elements. Example images are shown in Fig. 5. For color independence to occur, it is crucial that color variations neither alter the overall meaning of the image nor introduce significant ambiguity. This creates an ideal scenario in which concepts remain consistent across all subimages. However, it is essential to note that shared concepts do not always lead to correct predictions. For instance, in the attached car image, the true label was "auto showroom". Although all variations predicted a car-related class, only the original image produced the correct label.

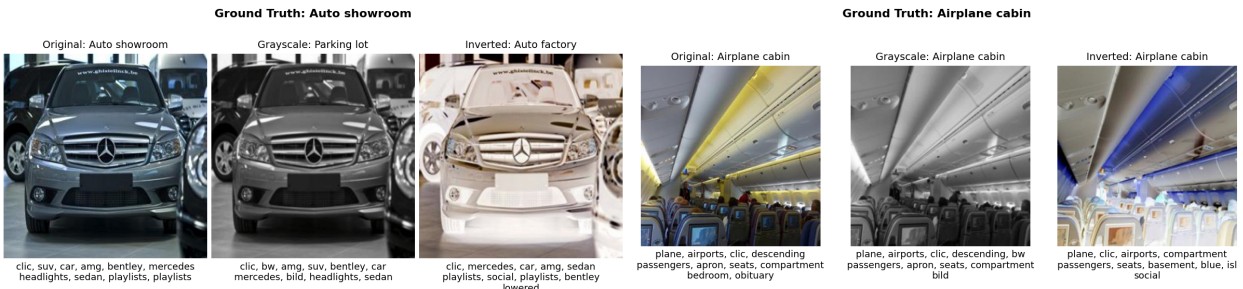

Figure 5: Example of two Color-Invariant Images with Top 10 concepts under images

The exact opposite can be said for color-dependent images. We have observed them slightly more often, with a total of 697 times, making up almost 10% of all sample data. This suggests that most images are in between significant color dependence and invariance. What characterized those images was that the image meaning was not entirely clear with the color change. Even a change to grayscale can introduce ambiguity, making it difficult to infer what the initial image was referring to clearly. An example can be seen in Fig 6.

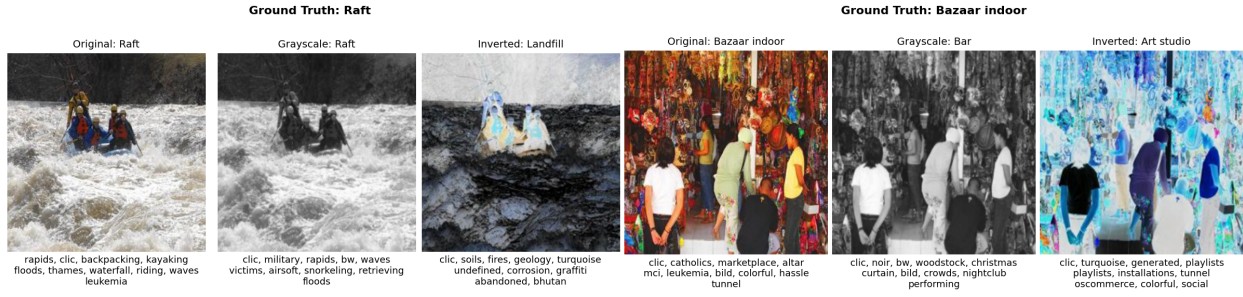

Figure 6: Example of Color Dependent Images with Top 10 concepts listed under images

In the provided "raft" example, the original image shares the concept of "rapids" with the grayscale image but otherwise tends to produce more activity-focused concepts, such as "kayaking" or "backpacking". In contrast, the grayscale image leans toward more serious themes, generating concepts like "military", "victims", and "airsoft". Finally, the inverted color map perturbs the image, making the resulting concepts seem entirely unrelated. Interestingly, even though the concepts significantly differ between the original and grayscale images, the predicted output is the same for both. On the other hand, the case of "Bazaar indoor" is only detected for the original image as the vibrant colors make it far more possible to interpret correctly.

These findings highlight that while the SAE method demonstrates an understanding of colors, it is not entirely color-independent, which was further supported by the perturbed validation accuracy. Specific images require color information to ensure successful recognition and accurate concept generation. This

dependency is particularly apparent in cases where the alteration of color introduces ambiguity, leading to shifts in the perceived meaning of the image. However, the ability of the SAE to maintain consistent concepts across a subset of images showcases its potential robustness to color, given a clear enough leading motive in the image.

### 4.2.2 Effect of reflection perturbations on concept representations

The results of the reflection modification experiment, shown in Table 4, reveal that while horizontal reflections had negligible impact, vertical reflections significantly affected the model's performance.

| Transformation | Original | Horizontal | Vertical |
|---|---|---|---|
| Accuracy (%) | 49.9 | 49.5 | 25.4 |

Table 4: Top-1 Validation Accuracy (%) comparison across transformations.

The decline in classification performance under vertical rotation can likely be attributed to the modified interpretation of visual context. When an image is flipped horizontally, objects' overall structure and interaction typically remain consistent. However, vertical flipping can introduce some unnatural scenarios. For instance, consider an image of a waterfall when flipped vertically, and the water appears to flow upward, which contradicts the typically observed representations of such scenes.

From examples in our data, we can observe these phenomena in practice. In Figure 7, we see an interesting case where a vertically flipped landscape image is misclassified as a pond. This is expected, as ponds often produce reflections resembling mirrored versions of their surrounding areas.

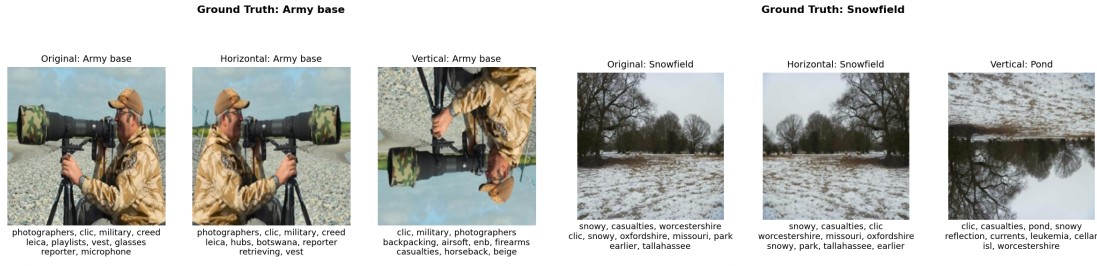

Figure 7: Examples of images with correct (left) and incorrect (right) classification when flipped vertically.

The insights from this section offer a deeper understanding of how concepts are represented and what types of spatial information the model relies on for prediction. The results demonstrate that while horizontal transformations have virtually no effect, vertical reflections significantly impair the model's performance, revealing some weaknesses in reliance on spatial patterns.

### 4.2.3 Finding spurious correlations and enhancing interpretability using LIME

As shown in Figure 11, the generated concepts often lack a clear relationship to the image. To better interpret these concepts, we apply LIME, a tool to identify the most influential features in a model's prediction. For image classification, LIME: 1) Segments the image into hyperpixels, 2) Perturbs segments and observes changes in label confidence, and 3) Identifies the hyperpixels that most influence the model's decision.

The yellow-highlighted areas visualize the regions with the most influence on each concept. For instance, the concept "quarterback" correctly corresponds to the player positioned in the back. Similarly, the concept "arch" aligns intuitively with the top portion of the rock formation. Interestingly, the origin of the concept "softball" in the third image is not immediately apparent. However, the LIME output reveals that its influence is focused on the bent-over individual, resembling a catcher in a softball game. This example underscores the utility of LIME in identifying potential spurious correlations.

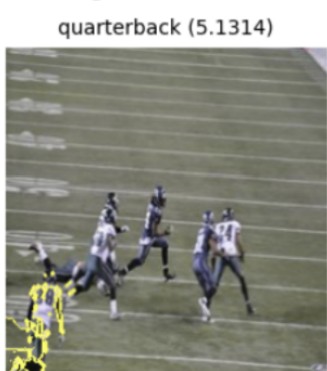
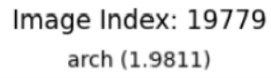
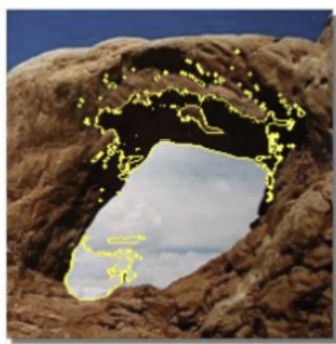
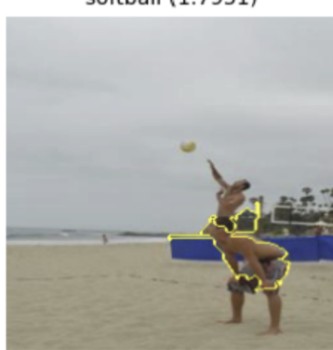

Figure 8: LIME explanation for high-activating concepts from the Places365 dataset.

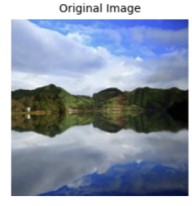
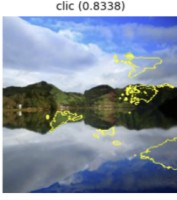
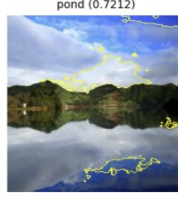
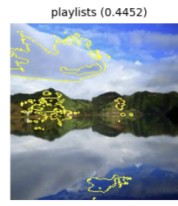
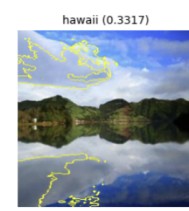
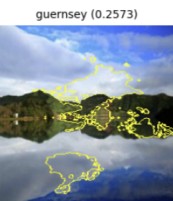

Figure 9: Explaining obscure top concepts using LIME.

We now revisit the examples from Figure 9 to analyze obscure top concepts, focusing specifically on the third image. Using LIME, we observe that the concept of "pond" is associated with the blue water and parts of the sky. The concept of "playlists" stems from a combination of hills, sky, and water, possibly reflecting common elements in playlist cover images. Interestingly, "hawaii" focuses entirely on the clouds, while "guernsey" primarily highlights the land. These insights illustrate how LIME helps reveal the specific image features influencing concept formation.

## 5   Conclusion

This study pursued a dual objective: first, to rigorously reproduce the key findings concerning Discover-then-Name Concept Bottleneck Models (DN-CBM) as presented by Rao et al. (2024), and second, to extend this work by critically evaluating the model's robustness and the depth of its interpretability under various perturbations. Our reproduction efforts essentially confirmed the original paper's central claims. We verified that the Sparse Autoencoder (SAE) effectively discovers potentially human-interpretable concepts from CLIP features, demonstrating the task-agnostic nature of this approach across CIFAR-10, CIFAR-100, Places365, and ImageNet datasets. Furthermore, we substantiated the claim that vocabulary granularity influences the precision of concept naming.

However, our investigation also highlighted nuances and potential limitations. While the generated concepts often aligned well with image content, enabling interpretable predictions via the linear probe, instances of obscure or seemingly irrelevant concept contributions persisted (Appendix A.2). Our user study indicated that while participants often preferred DN-CBM visualizations, the quantitative advantage in semantic consistency and name accuracy over raw CLIP features was statistically significant but visually less pronounced for well-aligned concepts, suggesting the interpretability claim requires careful qualification.

Our novel experiments focusing on robustness revealed specific vulnerabilities of the DN-CBM approach. The model exhibited marked sensitivity to color alterations, particularly color inversion, resulting in substantial accuracy drops and altered concept activations. This suggests an over-reliance on color cues for specific classification tasks, limiting robustness in scenarios with varying illumination or color styles. Similarly, while invariant to horizontal reflections, the model's performance degraded significantly under vertical reflections, indicating a reliance on canonical orientations rather than a more abstract understanding of scene composition. These findings indicate potential weaknesses in the model's generalization ability beyond learned visual statistics.

Integrating Local Interpretable Model-Agnostic Explanations (LIME) proved beneficial, extending the interpretability beyond simple concept lists. LIME allowed us to visually ground concept activations within image regions, confirming the origins of intuitive concepts and, crucially, shedding light on the features driving seemingly spurious or obscure concept activations (e.g., identifying specific textures or shapes correlated with unexpected concepts). Future research should focus on: 1) enhancing model robustness to visual variations via adversarial training or data augmentation; 2) improving concept naming to resolve ambiguities using larger vocabularies or contextual information; 3) integrating LIME during training for early identification of spurious correlations; and 4) validating the framework's scalability on more complex real-world data.

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

# A    Appendix

## A.1    Additional results for concept clustering

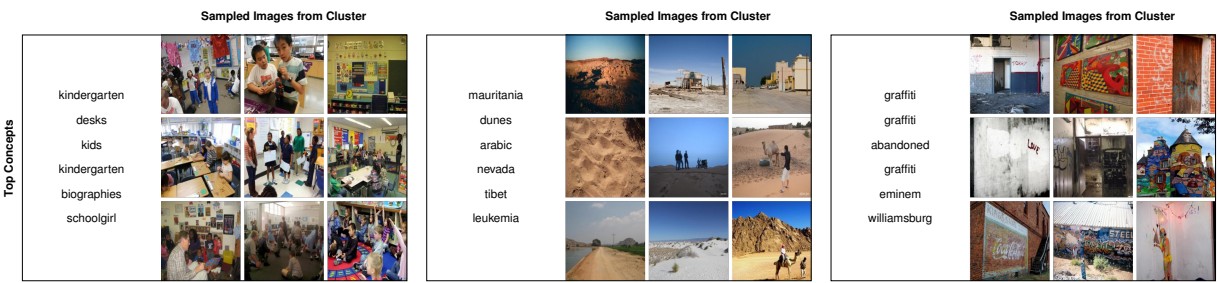

Figure 10: Concept clusters for Places365 with repeated concept names.

The results of the reflection modification experiment, presented in Table 4, indicate that horizontal reflections had a negligible impact on the model's performance, whereas vertical reflections significantly affected it.

## A.2    Additional results for local explanations

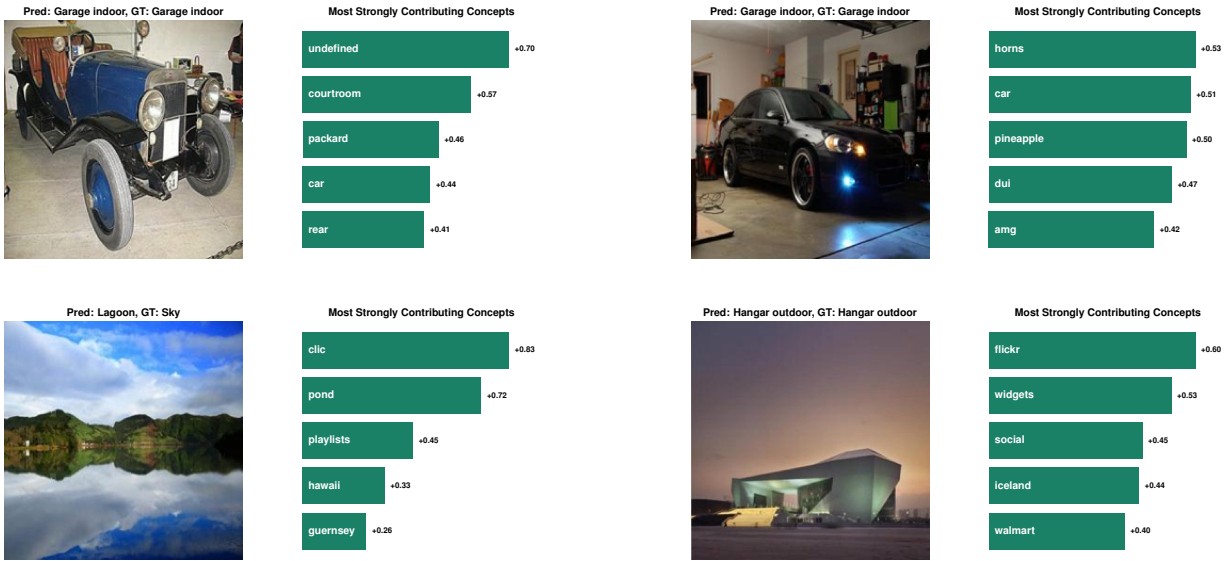

Figure 11: Local explanations for images with obscure top concepts contributions.

For instance, concepts like "undefined", "clic", "horns", and "flickr" emerged as top concepts for images demonstrably unrelated to them, raising concerns about the robustness of this method. In the top row of Figure 11, despite the clear presence of a car in both examples and the expectation for "car" to be the dominant concept with a high contribution score, its prominence is surprisingly low. Similarly, while the bottom left image features the relevant concept "pond", it is accompanied by other contributing concepts with tenuous connections to the image content. Finally, despite a correct prediction, the bottom right example offers a weak explanation tied possibly to "walmart" (associable with the building present), yet this link is tenuous, especially considering the lack of meaningful information provided by other top concepts.

### A.3   Additional visualizations for color augmentation experiment

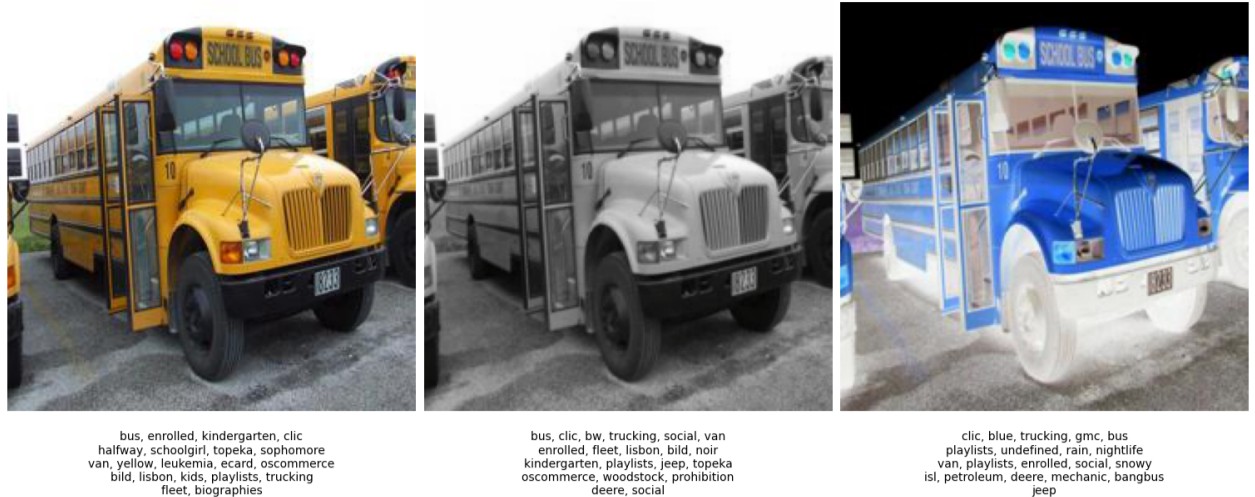

Figure 12: Top 20 concepts extracted for: 1. Original Image 2. Grayscale Image 3. Inverted Color Map Image

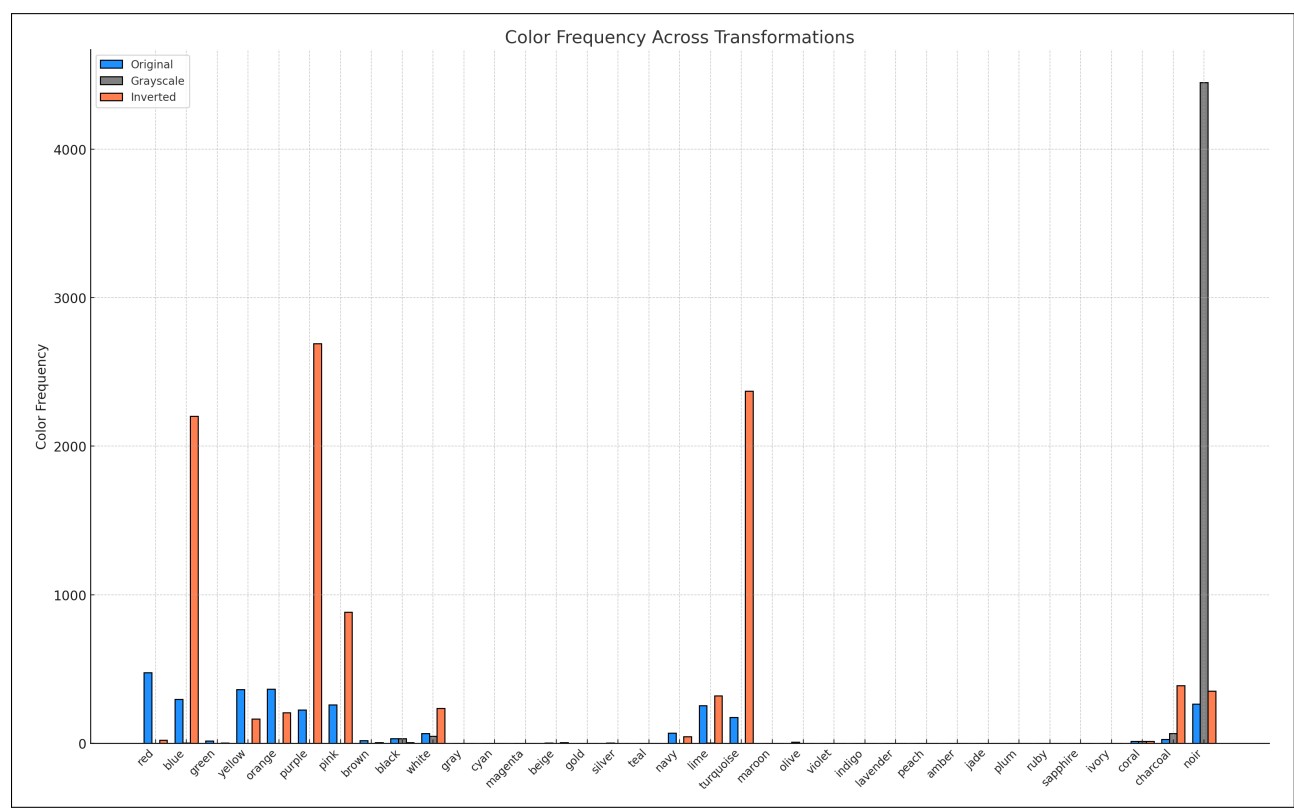

Figure 13: Quantity of colors observed in Top 20 concepts

| Class | Original Accuracy | Grayscale Accuracy | Difference |
|---|---|---|---|
| Swimming pool outdoor | 0.65 | 0.00 | 0.65 |
| Garage outdoor | 0.80 | 0.15 | 0.65 |
| Legislative chamber | 0.85 | 0.20 | 0.65 |
| Greenhouse indoor | 0.70 | 0.10 | 0.60 |
| Excavation | 0.85 | 0.25 | 0.60 |
| Hotel room | 0.60 | 0.60 | 0.00 |
| Cockpit | 0.95 | 0.95 | 0.00 |
| Boxing ring | 1.00 | 1.00 | 0.00 |
| Ice skating rink outdoor | 0.65 | 0.65 | 0.00 |
| Natural history museum | 0.70 | 0.70 | 0.00 |

Table 5: Accuracy Comparison for Classes with Largest Absolute Differences and No Changes between Original and Grayscale

