# OpenReview forum: "Discover-then-Name Revisited: Enhancing Concept Bottle- Neck Models Interpretability"
_TMLR — Rejected by TMLR_

### Review · Reviewer_uzEP · 2025-03-14

**Summary Of Contributions:**

Overall comments:

The paper lacks a clear motivation for revisiting DN-CBM and fails to rigorously substantiate its claims. It introduces analyses that are weakly justified and do little to improve either interpretability or robustness evaluation. Moreover, the methodology section is incomplete, the rigor of the user study is questionable, and the rationale for linking robustness evaluation with LIME is unconvincing. These shortcomings lead to the conclusion that the paper does not meet the publication standards.

Summary:

This paper aims to replicate and extend the DN-CBM framework, which focuses on automated concept discovery and interpretability in neural networks. The authors affirm DN-CBM’s capabilities in automated concept discovery, task-agnostic applicability, and improved vocabulary granularity, while also exposing limitations. In addition to replication, the paper presents experiments with color perturbations to assess concept robustness and employs LIME to trace the correspondence between image features and discovered concepts. The results indicate that DN-CBM suffers from issues of color-invariance, leading to potential misinterpretations, and that its automated naming mechanism occasionally produces semantically meaningless concepts.

**Audience:**

No

**Broader Impact Concerns:**

While the paper does not include a Broader Impact Statement, this omission is acceptable given the domain nature of the proposed method.

**Claims And Evidence:**

No

**Requested Changes:**

- Add a dedicated section that clearly articulates why revisiting DN-CBM is necessary.
- Discuss whether there are critical claims in the original work that require validation, inconsistencies that need to be addressed, or deeper insights that can be gained through replication.
- Emphasize the novelty or potential impact of any new insights gained through the reproduction study.
- Provide a detailed, critical analysis supported by experimental evidence or literature to explain why the results of DN-CBM remain inconclusive.
- Clearly explain the rationale behind merging robustness analysis (color variations) with the use of LIME for interpretability.
- Justify why LIME is necessary for evaluating robustness and how it contributes to the understanding of DN-CBM’s behavior under color perturbations.
- Discuss whether LIME is simply highlighting expected feature weighting or if it reveals novel insights into the model’s reliance on specific features.
- Clarify how these insights contribute to a deeper understanding of DN-CBM’s limitations.

**Strengths And Weaknesses:**

Strength:

The study offers a comprehensive reproducibility analysis of DN-CBM by verifying claims and integrating additional experiments that probe its limitations. Notably, the incorporation of color perturbation experiments and LIME-based feature tracing adds value by exposing potential weaknesses in the concept generation process and interpretability. Furthermore, the investigation into how vocabulary choices affect concept granularity and classification performance is insightful.

Major weakness:

1. The paper does not clearly articulate the necessity of reproducing DN-CBM. Reproducibility studies should either validate critical claims, reveal inconsistencies, or provide deeper insights into the original work’s limitations. However, the paper merely confirms DN-CBM’s general functionality without offering a compelling rationale for why this reproduction is needed.

2. The claim that DN-CBM is more interpretable than CLIP remains inconclusive, yet the authors do not critically examine or challenge this argument. The discussion lacks depth in questioning why DN-CBM should be considered superior and fails to provide a meaningful counter-analysis.

3. The study attempts to analyze DN-CBM’s robustness to color variations and simultaneously employs LIME for interpretability. However, there is no clear reasoning behind merging these two aspects. Why is LIME necessary for robustness evaluation? The connection appears arbitrary rather than a well-motivated research question. In light of this, the application of LIME seems to function as a sanity check rather than a novel interpretability approach. This makes the claim of “enhanced interpretability” questionable.

4. The authors claim that DN-CBM is less robust to color variations and that adding LIME enhances interpretability. However, there is no explanation of why LIME’s results support this claim. Is LIME merely filtering out weak features while emphasizing dominant ones? If so, this is expected behavior rather than a novel discovery. More importantly, does this insight genuinely contribute to understanding DN-CBM, or is it an artificial link created for the sake of additional analysis?

5. While the paper demonstrates that DN-CBM struggles with color variations, it does not explore why this happens. Is the model overly reliant on color-sensitive features? Does the concept extraction process fail to disentangle color-dependent attributes? Without a deeper investigation, this finding remains descriptive rather than explanatory.


Minor weakness:

1. The methodological contribution is insufficiently elaborated. Key details are not presented in the methods section but are instead scattered in the experimental setup. This structure is problematic because methodological novelties should be explicitly described, not implicitly assumed. Additionally, linking to a GitHub repository instead of clearly documenting implementation details in the paper is poor practice, as it forces readers to examine raw code instead of understanding the methodology from the paper itself.

2. The user study lacks rigor. The survey questions are not included, nor is there a discussion of how they were designed. To ensure meaningful user survey, the authors should refer to established frameworks in survey design, such as those in Tourangeau et al. (2000). Without this, the validity and reliability of the user study results remain questionable.

3. The paper fails to adhere to the double-blind review guidelines by openly sharing a GitHub repository without anonymization. This compromises the integrity of the review process and is a critical flaw.

---

### Review · Reviewer_ASka · 2025-03-15

**Summary Of Contributions:**

The paper reproduces the experiments of the paper Discover-then-Name by Rao et al. which proposed a method for sparsely encoding the features of a CLIP model and finding natural language embeddings that aligned with the sparse features. They confirm most of the findings in the original work, except that DN-CBM features are more interpretable than CLIP features. The paper then extends beyond the original paper in two sets of experiments. 1) They find that some concepts are highly susceptible to color changes. 2) They propose a method for assigning regions of the image to the concept.

**Audience:**

Yes

**Broader Impact Concerns:**

No broader concerns.

**Claims And Evidence:**

Yes

**Requested Changes:**

My suggested change is the following:
- The paper does not discuss related work for interpretability and concept bottlenecks, CLIP and Image-Text models, or sparse encodings. I think it would help readability to discuss this more in the introduction, add a related work, and place their contribution of the LIME technique in the context of previous methods like Grad-CAM.

This change is not critical to acceptance, but I think it would greatly improve the readability for people that are not familiar with this subfield.

**Strengths And Weaknesses:**

Strengths:
- The experiments are well done, and the reproduction is thorough. The experiments and methodology are well documented and the claims are well supported by the results. Overall the structure of the paper is reasonable to follow and the visualizations are illustrative. Overall I like the paper and appreciate the reproduction.


Weaknesses:
- I think a related work section would be useful for readers not as familiar with the literature. At least a brief discussion on interpretability, CLIP and other Image-Text models, and sparse encodings would help place this work in context particularly for the results that go beyond reproducing DN-CBM. Also it would be useful to discuss concept bottlenecks in general and cite the original paper Concept Bottleneck Models.

- Overall the experiments for reproduction of DN-CBM are thorough, but the extensions seem lacking a bit in experimentation. For example, there is no quantitative evaluation or comparison with other methods for LIME. For the color invariance analysis, the authors only try grayscale and inversion. It would be interesting to see more subtle color shifts or lighting changes.

- The introduction is a bit non-standard in that it is more like a table of contents, enumerating what each section contains. I think it would help put the paper in context to discuss Concept BottleNeck Models, mechanistic interpretability, and CLIP models. Also acronyms like SAE and LIME are thrown out without first saying what the acronym stands for. For someone not familiar with these concepts, it would be hard to understand.

- The LIME approach looks very similar to Grad-CAM and Shapley Values. If the authors want to claim LIME as a contribution they should place it in context and compare with similar approaches.

-The captions are a little uninformative. It would be nice if they were self-contained.

---

### Review · Reviewer_UGwd · 2025-03-31

**Summary Of Contributions:**

The work under review aims to replicate and extend the research on Discover-Then-Name Task-Agnostic Concept Bottlenecks via Automated Concept Discovery introduced by Rao et al. (2024). The authors provide experiments, providing evidence that that the majority of the original claims hold true, with a few exceptions: the trained SAEs are shown to generate a latent concept space that is interpretable by humans and most concepts receive meaningful names.The paper also presents additional experiments that involve modifying the colors in the input images and examining the resultant changes in concepts.

**Audience:**

No

**Claims And Evidence:**

No

**Requested Changes:**

To strengthen the paper, in addition to resolving the weaknesses listed in the "Strengths And Weaknesses" section, it would be beneficial for authors to consider conducting additional experiments that differ from those in the main paper by Rao et al. (2024). For instance, the authors might compare the performance of SAEs extracted concepts with standard neurons or linear extraction methods such as NMF [3] or clustering approaches. Moreover, the process of naming concepts could be evaluated against alternative methods (e.g., as presented in [1], [4], [5]).

[3] Fel, Thomas, et al. "Craft: Concept recursive activation factorization for explainability." Proceedings of the IEEE/CVF Conference on Computer Vision and Pattern Recognition. 2023.

[4] Bykov, Kirill, et al. "Labeling neural representations with inverse recognition." Advances in Neural Information Processing Systems 36 (2023): 24804-24828.

[5] Hernandez, Evan, et al. "Natural language descriptions of deep visual features." International Conference on Learning Representations. 2021.

**Strengths And Weaknesses:**

### Strengths:
1. Replication studies contribute positively to the field.
2. The paper is written in a manner that is easy to follow.

### Weaknesses:

1. While conducting the replication study, the paper offers few novel insights beyond a minor experiment involving color disturbance.
2. Many of the experiments are qualitative, which makes it challenging to rigorously assess the claims.
3. The section on the effectiveness of concept interventions from the original paper by Rao et al. (2024) is not reproduced.
4. The discussion on enhancing classification decision interpretability with LIME is predominantly qualitative and lacks quantitative evaluation.
5. The description of the user study experiment is incomplete; for example, the details regarding its setup and statistical significance of the results are missing.
6. Furthermore, the assertion that the extracted concepts are interpretable is solely supported by the user study. This assessment could be augmented with quantitative measures—for example, by evaluating the performance of the concepts in a binary classification setting, determining how effectively a concept can recognize images corresponding to the same label [1], or through generative approaches [2].

### Minor weaknesses:

1. The GitHub link includes what appears to be the name of the author. To adhere to the double-blind review format, such links should be omitted from the submission.
2. On page 4, the reference "We use the Python library lime" is broken.

[1] Oikarinen, Tuomas, and Tsui-Wei Weng. "Clip-dissect: Automatic description of neuron representations in deep vision networks." arXiv preprint arXiv:2204.10965 (2022).

[2] Kopf, Laura, et al. "Cosy: Evaluating textual explanations of neurons." Advances in Neural Information Processing Systems 37 (2024): 34656-34685.

---

### Decision · Action_Editor_CTWm · 2025-05-02

**Recommendation:** Reject

**Comment:**

Reviewers have unanimously recommended rejection primarily due to a lack of engagement from the authors in responding to the queries raised. In general, reviewers aligned on the areas of concern and the weaknesses identified. Key issues included a lack of novelty in the contributions beyond replicating the original work, particularly regarding the claims in the abstract about going beyond replication, insufficient experimental evidence to support the claims of enhanced interpretability and robustness, as well as an incomplete methodology section. Additionally, the rationale for integrating LIME with robustness evaluation was unconvincing. I would encourage the authors to take the feedback into account and further improve their manuscript, and subsequently consider resubmitting for consideration.

**Audience:**

The intended audience for this paper includes researchers in the fields of machine learning, artificial intelligence, and specifically those focused on model interpretability and explainability. The paper is particularly relevant to individuals interested in concept bottleneck models, automated concept discovery, and the application of sparse autoencoders. It also targets those looking to understand the robustness and interpretability of neural networks under various "feature" perturbations. The experimental setup and analysis make it suitable for a broader AI audience and beyond, including individuals involved in reproducibility studies and developing interpretable AI models.

**Claims And Evidence:**

This submission entitled "Discover-then-Name Revisited: Enhancing Concept Bottleneck Models Interpretability" aims to replicate and partially extend the Discover-Then-Name Concept Bottleneck Models (DN-CBM) introduced by Rao et al. (2024). The authors claim that DN-CBM enhances traditional CBMs by utilising sparse autoencoders for automated concept discovery, leading to improved interpretability and task-agnostic applicability. They validate these claims through experiments on datasets like CIFAR-10, CIFAR-100, Places365, and ImageNet, confirming automated concept discovery and the impact of vocabulary on concept granularity. However, the claim of improved interpretability over CLIP remains inconclusive, according to the authors of this TMLR submission. Additional experiments reveal limitations in the model's robustness to colour variations and the effectiveness of LIME in enhancing interpretability. Reviewers noted that while the replication of the original experiments was thorough, the novel contributions lacked sufficient experimental evidence and critical analysis, particularly regarding the interpretability and robustness claims.

**Resubmission Of Major Revision:**

The authors may consider submitting a major revision at a later time.